# Next-Generation Sequencing of the Human Aqueous Humour Microbiome

**DOI:** 10.3390/ijms25116128

**Published:** 2024-06-01

**Authors:** Günther Schlunck, Philip Maier, Barbara Maier, Wolfgang Maier, Sebastian Strempel, Thomas Reinhard, Sonja Heinzelmann

**Affiliations:** 1Eye Center, Medical Center, Faculty of Medicine, University of Freiburg, 79110 Freiburg im Breisgau, Germany; guenther.schlunck@uniklinik-freiburg.de (G.S.); philip.maier@uniklinik-freiburg.de (P.M.); thomas.reinhard@uniklinik-freiburg.de (T.R.); 2Institute for Infection Prevention and Control, Faculty of Medicine, University of Freiburg, 79110 Freiburg im Breisgau, Germany; barbara.maier@uniklinik-freiburg.de; 3Bioinformatics Group, Department of Computer Science, University of Freiburg, Georges-Köhler-Allee 106, 79110 Freiburg im Breisgau, Germany; maierw@informatik.uni-freiburg.de; 4Microsynth AG, Schützenstrasse 15, CH-9436 Balgach, Switzerland; sebastian.strempel@microsynth.ch

**Keywords:** microbiome, eye, anterior chamber fluid, next-generation sequencing

## Abstract

The microbiome of the ocular surface has been characterised, but only limited information is available on a possible silent intraocular microbial colonisation in normal eyes. Therefore, we performed next-generation sequencing (NGS) of 16S rDNA genes in the aqueous humour. The aqueous humour was sampled from three patients during cataract surgery. Air swabs, conjunctival swabs from patients as well as from healthy donors served as controls. Following DNA extraction, the V3 and V4 hypervariable regions of the 16S rDNA gene were amplified and sequenced followed by denoising. The resulting Amplicon Sequence Variants were matched to a subset of the Ribosomal Database Project 16S database. The deduced bacterial community was then statistically analysed. The DNA content in all samples was low (0–1.49 ng/µL) but sufficient for analysis. The main phyla in the samples were *Acinetobacteria* (48%), *Proteobacteria* (26%), *Firmicutes* (14%), *Acidobacteria* (8%), and *Bacteroidetes* (2%). Patients’ conjunctival control samples and anterior chamber fluid showed similar patterns of bacterial species containing many waterborne species. Non-disinfected samples showed a different bacterial spectrum than the air swab samples. The data confirm the existence of an ocular surface microbiome. Meanwhile, a distinct intraocular microbiome was not discernible from the background, suggesting the absence of an intraocular microbiome in normal eyes.

## 1. Introduction

The human body is colonised by an abundance of microbial cells, which have a significant role in health and disease. Next-generation sequencing techniques allow us to characterise the microbiomes of various body sites, including the skin, oral, nasal, conjunctival, vaginal, and gut mucosal microbiomes, in health as well as in distinct disease states, e.g., in inflammatory bowel disease, pre-diabetes, or preterm birth [1,2].

Conflicting results have been reported in samples from body sites commonly considered free of bacteria, such as cerebrospinal fluid or placental tissue. A placental microbiome has been described [3], whereas another study argued against a viable microbial colonisation of the human placenta [4].

Using 16S-PCR and short-read sequencing, bacterial DNA can be detected with high sensitivity and specificity. This technique also allows for detecting microbes not amenable to cultivation. However, the presence of bacterial DNA does not necessarily indicate the existence of living bacteria as bacterial remnants can be stable for longer periods and phagocytic or infected cells may carry bacterial DNA to distant sites [5]. Furthermore, the ubiquitous presence of microbes in the environment and the extremely high sensitivity of PCR raises the risk to detect contaminating DNA and mistaking the result as indicating resident microbial colonisation [5]. This issue is of particular importance when body sites with a very low bacterial load, such as the ocular surface, are examined. Thus, particular care must be taken to control for the presence of background bacterial DNA in these analyses.

The microbial load of skin or oral mucosa (12 bacteria per epithelial cell) is about 200 times higher than that of conjunctival mucosa (0.06 bacteria per epithelial cell [6]. The microbiome of the healthy ocular surface has been characterised [7], and the current literature suggests that several factors influence the microbial patterns detected on the ocular surface, namely, the sampling method [8,9], the timing of sampling, the ethnicity, age, and sex of the sampled individual [10,11], and the sampling location (upper or lower fornix, limbus, cornea) [12,13].

Today, 16S rDNA gene sequencing facilitates insights not only into a healthy but also a diseased ocular surface, which would never have been possible with bacterial culture methods [14].

Under healthy conditions, living bacteria should be prevented from passing through major barriers such as the blood–brain, blood–placenta [15,16], or blood–retina barrier [17]. However, it has been reported that these barriers may be breached under certain conditions [18,19]. Currently, little is known about a possible physiological presence of microbes in healthy eyes. In one study comparing vitreous samples from patients with or without bacterial endophthalmitis, no bacterial DNA was retrieved from healthy eyes by whole-genome sequencing [19], but so far no comparable study has been performed with the more sensitive 16S rDNA sequencing. Another study did not find an intraocular microbiome in the aqueous humour of pseudophakic donor eyes using shotgun sequencing but detected contaminating environmental DNA [20]. If healthy eyes harbour resident microbes, it is important to understand their role in immunological pathomechanisms and anterior-chamber-associated immunodeficiency (ACAID) [21,22]. Accordingly, we used 16S rDNA gene sequencing and employed several controls to detect potential contaminants.

## 2. Results

The patients’ and controls’ demographic data are displayed in Table 1.

Following isolation, amplification, and filtering of the DNA, 1,107,324 reads were obtained in total (median 26,787 +/− 7315 per sample). The mean read length per sample was 236 +/− 12 base pairs. Description of the analysis quality via FastQC (version 0.11.8) revealed acceptable parameters despite the low content of DNA per sample (0–1.49 ng/µL). Classification of the bacterial DNA showed *Actinobacteria* (48%), *Proteobacteria* (26%), *Firmicutes* (14%), *Acidobacteria* (8%), and *Bacteroidetes* (2%) as the main phyla in all samples (Figure 1).

To display differences between the samples, a detrended correspondence Analysis (DCA) and a sample correlation diagram were used (Figure 2a,b).

DCA revealed clustering of all samples related to the surgical patients, including the corresponding dry swab negative controls. These “surgical” samples contained mainly *Acinetobacteria*, classified as *Arthrobacter russicus* (see Figure 1 left side), and waterborne bacteria (e.g., *Pseudomonas* and *Sphingomonas* species), which are typically not dominant in published reports on the conjunctival microbiome. The positive controls of two healthy eyes (“native” samples) not undergoing cataract surgery were clearly distinct from a corresponding air swab negative control. These positive and negative controls strongly differed from the cluster of “surgical” samples (Figure 2a,b). The sample correlation diagram shows the correlation coefficients of the samples (Figure 2b).

Correlations of ASV abundance for all 12 samples (“Air”, “GS”, and “SH”) show the least correlation in comparison to the intraoperatively obtained samples (DS (dry swab), CS (conjunctival swab) and AC (anterior chamber fluid), 1–3 each, numbered top down). Between the intraoperative categories, specific correlation is detectable as well: the probes from different patients differ among each other. Anterior chamber fluid 1 and dry swab 1 show the most different bacterial DNA of all “intraoperative” samples.

## 3. Discussion

This study aimed to explore a possible microbiomic signature of the human aqueous humour by next-generation 16S rDNA gene sequencing.

As this method is highly sensitive, care was taken to characterise environmental background DNA using several controls. The two positive control conjunctival swabs from untreated conjunctiva clearly differed from a corresponding air swab, pointing to the presence of a local microbiome on an untreated ocular surface, as reported previously [8,13,23]. Unexpected was the clustering of all surgery-related samples including intraocular aspirates, conjunctival swabs, and air swab negative controls (Figure 2), which were in marked contrast to the positive and negative controls unrelated to surgery, as mentioned above. This strongly suggests an effect of preoperative disinfection and irrigation, which appear to clear bacterial DNA from the ocular surface (see also [24]). This could explain the similarity of conjunctival samples from surgical patients to the corresponding dry air swabs. A clustering of aqueous humour samples with the former two groups is consistent with the absence of a distinct intraocular microbiome. *Arthrobacter* species are frequently isolated from environmental sources including the air of ventilation systems [25] and were detected in high abundance in all samples related to surgical patients. To date, no cultivation of *Arthrobacter russicus* from the ocular surface has been reported. As a consequence, environmental contamination seems to be the most plausible explanation for the high abundance in our samples. Intriguingly, the difference in the surgery-related air swabs and the positive-control-related air swabs obtained on a different date but in the same operating theatre indicates that the method is sensitive enough to detect differences in background contamination, which were more pronounced than the differences detected between the surgery-related samples (air swab, cleaned conjunctiva, aqueous humour) on a single day.

In general, the detection of bacterial DNA by 16S rDNA gene sequencing does not allow for a statement about living bacteria. It is possible that sterile surgical instruments and flushing solutions like basal salt solution contain bacterial DNA introduced in the production and sterilisation processes, typically waterborne species like in our surgery-related samples. If the DNA content of the sampled material of interest is very low, this background DNA may become dominant due to the high sensitivity of the detection system, giving a poor signal-to-noise ratio.

In the future, with more samples from healthy and diseased eyes, together with appropriate controls, more distinct statements about an intraocular microbiome in health and disease should be possible, with implications for clinical use.

## 4. Materials and Methods

This study followed the rules of the Declaration of Helsinki of 1975, revised in 2013. It was approved by the local ethics committee (23/20) and is listed in “Deutsches Register Klinischer Studien” (DRKS00020512). All subjects gave their informed consent for inclusion before they participated in this study.

### 4.1. Patients

In total, three Caucasian patients undergoing cataract surgery by the same surgeon on the same day in the same operation theatre were included. There were two inclusion criteria: informed consent and the absence of acute or chronic eye disease besides a cataract at the time of surgery. Exclusion criteria were local (eye drops/ointment) or systemic antibiotic therapy during the last 6 months, contact lens wearing, history of infectious or noninfectious inflammatory eye disease in the past, known allergy to any of the drugs for surgery (see ‘sampling’ below), diabetes mellitus, and chronic inflammatory bowel disease. Two other Caucasian individuals who met the inclusion and exclusion criteria but without a cataract and planned cataract surgery served as conjunctival and air sampling controls. These two samples were taken together on the same day in one operation theatre, but on a different date to the three cataract patients’ samples.

### 4.2. Sampling

All three cataract patients underwent routine cataract surgery. For topical anaesthesia, 2% xylocaine and 0.5% proparacaine were applied to the ocular surface three times every five minutes. Subsequently, 5% povidone-iodine was applied to the ocular surface, superior and inferior fornices, and the periocular lid region. After 3 min, a sterile cover was placed on the face and eyelids, a lid retractor was inserted, and the povidone-iodine was flushed from the conjunctiva by balanced salt solution (BSS, Alcon, Freiburg im Breisgau, Germany). From each patient, three samples were taken by one surgeon wearing sterile gloves and clothes: first, a dry swab (Sugi^®^ Eyespear pointed tip, Questalpha, Eschenburg, Germany) was waved in the air above the ocular surface and then placed in a 2 mL sterile microtube (Sarstedt, Nümbrecht, Germany). Second, another dry swab was used to wipe the limbal conjunctiva at the site of the future incision and was also placed in a microtube. Third, just after the initial corneal incision, 60–100 µL of anterior chamber fluid was aspirated with a 1 mL syringe, and the injection needle and the collected fluid were placed in a microtube. The samples were put on dry ice immediately and prepared for shipment. For the two conjunctival swab controls, sampling was performed using a first dry swab waved in ambient air and a second swab used to wipe the inferior fornix (without any pretreatment). The swabs were transferred to microtubes, placed on dry ice, and prepared for shipment. Individual sterile gloves were used for each sample. All samples were sent to the sequencing service provider on dry ice and processed there without any additional storage (Microsynth AG, Balgach, Switzerland). Processing included individual negative and positive controls during each step.

### 4.3. Sample Processing

Extraction, lysis, and DNA isolation were performed according to the manufacturer’s recommendations (ZymoBIOMICS DNA Mini Kit, Freiburg im Breisgau, Germany). Bead beating was run on a FastPrep-24 instrument (MPBiomedicals, Irvine, CA, USA; 4 cycles of 45 s at speed 4 followed by 1 cycle of 45 s at speed 6.5). We prepared 400 µL of raw extract for DNA isolation. The concentration of the isolated DNA was assessed with PicoGreen measurement (Quant-iT™ PicoGreen™ dsDNA Assay Kit, Thermo Fisher, Waltham, MA, USA).

To sequence the V3 and V4 regions of the bacterial 16S rDNA gene, two-step, Nextera barcoded PCR libraries using the locus-specific primer pair 341F (*5′-CCT ACG GGN GGC WGC AG-3′*) and 805R (*5′-GAC TAC HVG GGT ATC TAA TCC-3′*) with 20 PCR cycles for the first step and 20 PCR cycles for the second step were created. Subsequently, the libraries were sequenced on an Illumina MiSeq platform (Illumina, San Diego, CA, USA) using a v2 500 cycles kit.

### 4.4. Biostatistics

The produced paired-end reads that passed Illumina’s chastity filter were subject to de-multiplexing and trimming of Illumina adaptor residuals using Illumina’s bcl2fastq software version v2.20.0.422. The quality of the reads was checked with the software FastQC version 0.11.8, and sequencing reads that fell below an average Q-score of 20 or had any uncalled bases (N) were removed from further analysis. The locus-specific V3 and V4 primers were trimmed from the sequencing reads with the software cutadapt v3.2. Paired-end reads were discarded if the primer could not be trimmed. Trimmed forward and reverse reads of each paired-end read were merged to reform the sequenced molecule in silico considering a minimum overlap of 15 bases using the software USEARCH version 11.0.667. Merged reads that contained ambiguous bases or were outliers regarding the expected amplicon size distribution were also discarded. The remaining reads were denoised using the UNOISE algorithm implemented in USEARCH to form ASVs (Amplicon Sequence Variants), discarding singletons and chimeras in the process. The resulting ASV abundance table was then filtered for possible barcode bleed-in contaminations using the UNCROSS algorithm. ASV sequences were compared to the reference sequences of the RDP 16S database provided by https://www.drive5.com/usearch/manual/sintax_downloads.html (accessed on 8 July 2020), and taxonomies were predicted considering a minimum confidence threshold of 0.5 using the SINTAX algorithm implemented in USEARCH. The microbial taxa that we found were visualised via krona charts. Alpha diversity was estimated using the Richness (Observed), Simpson, and Shannon indices (see Appendix A). Rarefaction analysis was carried out to estimate the coverage of the captured metagenome in contrast to the potential metagenome. Beta diversity was calculated using the weighted Unifrac distance method on the basis of rarefied ASV abundance counts per sample (see Appendix A). These sample distances were then used in a detrended correspondence analysis (DCA) to reveal possible patterns of inter-sample relations. Alpha and beta diversity calculations and the rarefaction analysis were performed with the R software packages phyloseq v1.26.1 and vegan v2.5-5. To detect differentially abundant ASVs depending on collected sample metadata (e.g., sample category—air, control, etc.), differential ASV analysis using normalised abundance counts was performed with the R software package DESeq2 v1.26.0.

## 5. Conclusions

In conclusion, these data are in line with other observations in ocular [19,20], placental [26], and cerebrospinal [27] material. The findings strongly suggest that there usually are no significant amounts of living bacteria in body sites shielded by blood–tissue barriers. With the next-generation sequencing methods currently available, no intraocular microbiome has been detected.

## Figures and Tables

**Figure 1 ijms-25-06128-f001:**
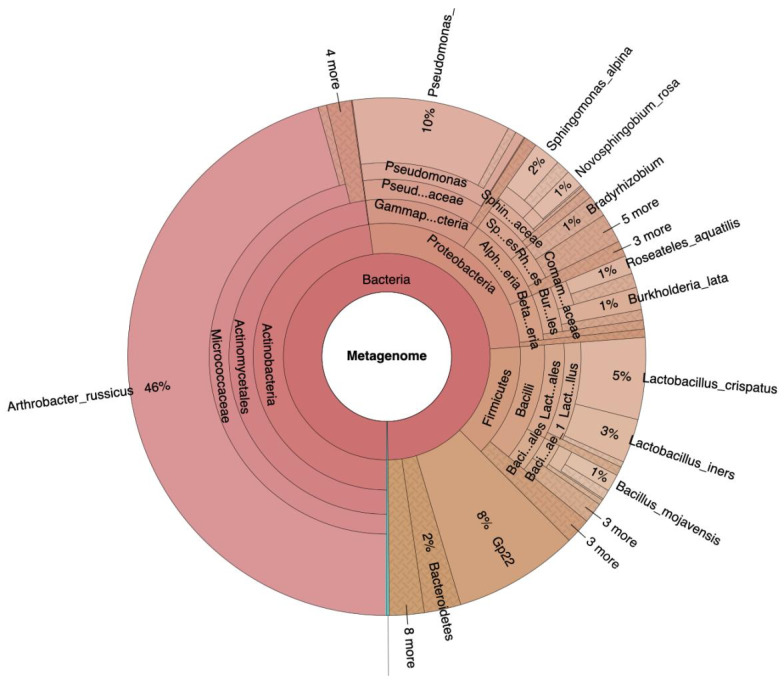
Krona chart of the entire microbial taxa of all samples: the main phyla are *Firmicutes*, *Actinobacteria*, *Proteobacteria*, and *Bacteroidetes*.

**Figure 2 ijms-25-06128-f002:**
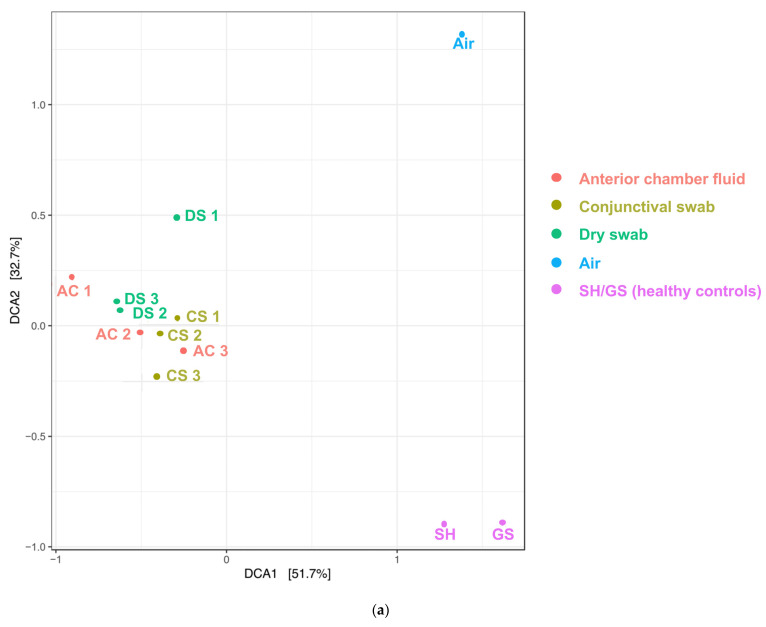
(**a**) Detrended correspondence analysis (DCA) using the normalised ASV abundances of the sequenced samples. Most of the observed abundance differences in this study are explained by differences between cataract patient samples and untreated controls (DCA1), and between negative and positive untreated controls (DCA2), but not by differences between the types of cataract patient samples. (**b**) Sample correlation diagram. Abbreviations: AC (anterior chamber fluid), CS (conjunctival swab, DS (dry swab), GS/SH (healthy controls)

**Table 1 ijms-25-06128-t001:** Demographics of the patients and control subjects.

	Age (years)	Sex	Ethnicity	Eye
Patient 1	70	Female	Caucasian	Left
Patient 2	73	Male	Caucasian	Right
Patient 3	83	Female	Caucasian	Right
Control 1	56	Male	Caucasian	Right
Control 2	43	Female	Caucasian	Right

## Data Availability

Due to ethical and safety reasons, the original data are unavailable for the public.

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
