# Peer review of "Next-Generation Sequencing of the Human Aqueous Humour Microbiome"

_ijms, 2024, doi:10.3390/ijms25116128_

Round 1

Reviewer 1 Report

Comments and Suggestions for Authors

Thank you for the submission of your manuscript entitled “Next-generation sequencing of the human aqueous humor microbiome”. The goal of this study was to investigate the presence of an intraocular (aqueous humor) bacterial microbiome using NGS. The investigators did an excellent job describing the use of negative controls to account for the possibility of environmental contaminants in their low biomass samples. Although the sample size is relatively small and not powered for statistical significance, the study is informative and aligns with previously published work with similar experimental designs. Please see below for revisions that are recommended before publication. 

Abstract:

Line 22: Information on DNA content of the sample is provided in the abstract but not the results. Please include this information in the results or supplementary materials.

Materials & Methods:

Line 101: What type of dry swabs were used for sample collection? Please include the manufacturer’s information. 

Line 111: Following shipment, were the samples stored prior to processing? If so, please include this information.

Lines 112 – 123: It is imperative to use negative controls at each step of the analysis in this field of study, as the authors mention low biomass environments are more susceptible to having contaminating DNA affect the results. The investigators do an excellent job including environmental controls (such as unused swabs), however; there is no mention of DNA extraction blanks (reagents only), no-template amplification reactions, and reagents-only sequencing samples. Negative controls should be included at every step, amplified, and sequenced alongside the study samples. Subtractive filtering should then be employed to ensure potential reagent and environmental contaminants do not confound the interpretation of microbiome data. It is not clear if extraction, amplification, or sequencing controls were included for contaminant filtering analysis to discount the possibility of reagent contaminants impacting the results. If not, this should be listed as a limitation to the study.

Line 142: The term “metagenome” is often used with shotgun sequencing in which all genomic DNA is read in a sample. This differs from 16S rRNA sequencing in which one specific region of DNA is read. Please confirm whether metagenome is an appropriate word to use for these methods and in Figure 1.

Lines 143-146: Please provide the results of the alpha diversity analyses (Observed ASVs, Simpson, Shannon) and beta diversity analyses (weighted Unifrac) in the results or supplementary materials.

Results:

Figure 2 a,b: The font size of the datapoints and sample labels is too small and difficult to read. 

Author Response

Dear reviewer, with great thankfulness we read and implemented your valuable comments. Here is our point-to-point reply. In case we misunderstood something please let us know (especially the presentation of the beta diversity is hopefully what you were looking for.....).  

Reviewer 2 Report

Comments and Suggestions for Authors

Schlunck et al present a research paper for next-generation sequencing of the human aqueous humor microbiome and the authors are appreciated for research planning, plan execution, data collection, and writing and presenting the research work as a research paper to publish in the journal and reaching a conclusion that the absence of an intraocular microbiome in normal eyes. The described work of this research paper seems original because few research papers are published on this topic using different research methodologies and approaches. However, it is not very clear after reading this research paper how the research approach described in this research paper is improved and better than other published papers. For example; Authors state that To date cultivation of Arthrobacter russicus from the ocular surface has not been reported. What is the difference between this published research paper (https://www.mdpi.com/2076-0817/10/4/405) and the author´s paper. How can the author state that the research approach described in the paper is better than this already published paper? The interest is to explain why a reader should read this research paper and be convinced to use this research approach and data in a clinical setup with patients. Overall, this research paper is good and acceptable for publication.

Author Response

Dear reviewer, find attached our comments to your ideas. Please let us know if we misunderstood your findings and proposals. Thank you so much for making us rethink our hypotheses!
